# A Percutaneous Catheter Solution as a Spacer for Regurgitant Heart Valve Disease

**Min-Ku Chon [1], Dong-Hoon Shin [2], Su-Jin Jung [3], Hyeon-Jun Park [3] and June-Hong Kim [1,***

[1] Cardiovascular Center, Department of Cardiology, School of Medicine, Yangsan Hospital, Pusan National University, Yangsan 50612, Korea; chonmingu@gmail.com
[2] Department of Pathology, School of Medicine, Yangsan Hospital, Pusan National University, Yangsan 50612, Korea; donghshin@chol.com
[3] R&D Center, Tau-PNU MEDICAL Co., Ltd., Yangsan 50612, Korea; jsj@tau-pnu.co.kr (S.-J.J.); hjpark@tau-pnu.co.kr (H.-J.P.)
* Correspondence: junehongk@gmail.com

**Abstract:** Spacers, such as FORMA (Edwards Lifescience) or Mitraspacer, are used to treat mitral regurgitation or tricuspid regurgitations. However, they require external therapeutic liquid filler injection into the spacer device. This should be leak-tight over the time of implantation, which is a major limitation in device design. Here, we propose a self-expandable spacer with a nitinol inner mesh and expanded poly (tetrafluoroethylene) (ePTFE) coating that also functions as a spacer. We designed nitinol 3D mesh templates, coated with a commercially available low and high durometer ePTFE membrane. Finally, we implanted the spacer into a swine pulmonary artery and right atrium (superior vena cava) as an intervention technique. Twenty-four swine were used, except in two cases suspected of procedural infection. The results were analyzed in the remaining 22 cases and all devices were easily delivered and had good function in self-expansion and implantation. After eight weeks, all individuals were examined for gross and pathological analysis to determine the biological safety of the device. There was no evidence of damage or other abnormalities and increased postoperative endothelialization outside of ePTFE coatings. In conclusion, this study suggests using a self-expandable spacer to complement the medical limitations of the existing filling-type spacer devices.

**Keywords:** biocompatible coating; ePTFE; cardiology; medical device; implantable device





## 1. Introduction

A spacer device has been used to block spaces or lesions in the heart, such as the regurgitant orifice in the conventional way of a catheter-based cardiac interventional treatment (e.g., FORMA by Edward Lifesciences, CA, USA [1–5], Percu-Pro™ System [6] and Mitra-Spacer™ by Cardiosolutions Inc., Stoughton, MA, USA [7]). The spacer is placed in the regurgitant orifice and creates a new surface for leaflet coaptation, which aims to assist coaptation of dilated valve leaflets. These spacers are designed to expand into suitable spatial fit, shape, and size inside of the heart for the purpose of the operator.

Air, saline, or other biocompatible fillers are injected from outside the body via a separate pathway ("lumen") into the spacer device. Technically, the fillers' physical properties affect the size, shape, and function of the device [8,9]. Furthermore, there is always a potential for filler leakage inside the body, so biocompatibility between the body and the fillers should be a priority for clinicians.

To overcome these limitations, we sought to propose a new type of spacer design that deviates from the filler concept. Applying a self-expandable type of Nitinol mesh does not require a separate material to fill the spacer for maintaining its shape. Structurally, the exterior of the spacer in direct contact with the tissue was coated with ePTFE (expanded-Poly tetra-fluoroethylene), a biocompatible material, to prevent foreign body reactions with the valve and the surrounding heart tissue.

This study evaluated the biocompatibility of the ePTFE-coated device by inserting it into the free space of the swine heart for 6–8 weeks. Consequently, we demonstrated a self-expandable spacer in the percutaneous devices to overcome the limitations of the existing filling-type spacer devices.

## 2. Materials and Methods

### 2.1. Spacer Design

2.1.1. Self-Expandable Mesh Design

We implemented a self-expandable mesh design using nitinol metal wire and made a spacer to unfold and maintain its shape without fillers. Figure 1a shows the spacer center that is bulging in one direction. The proximal ends of the mesh are connected to the tube, and the distal end of the mesh is freely moved along the tube. It is possible to load it into the delivery sheath (Figure 1b). It also allows blood to flow into the mesh through the sleeve created between the distal end of the mesh and the tube, thereby spreading the mesh to form itself. In addition to the sleeve, a spatial opening design can be added to the spacer.

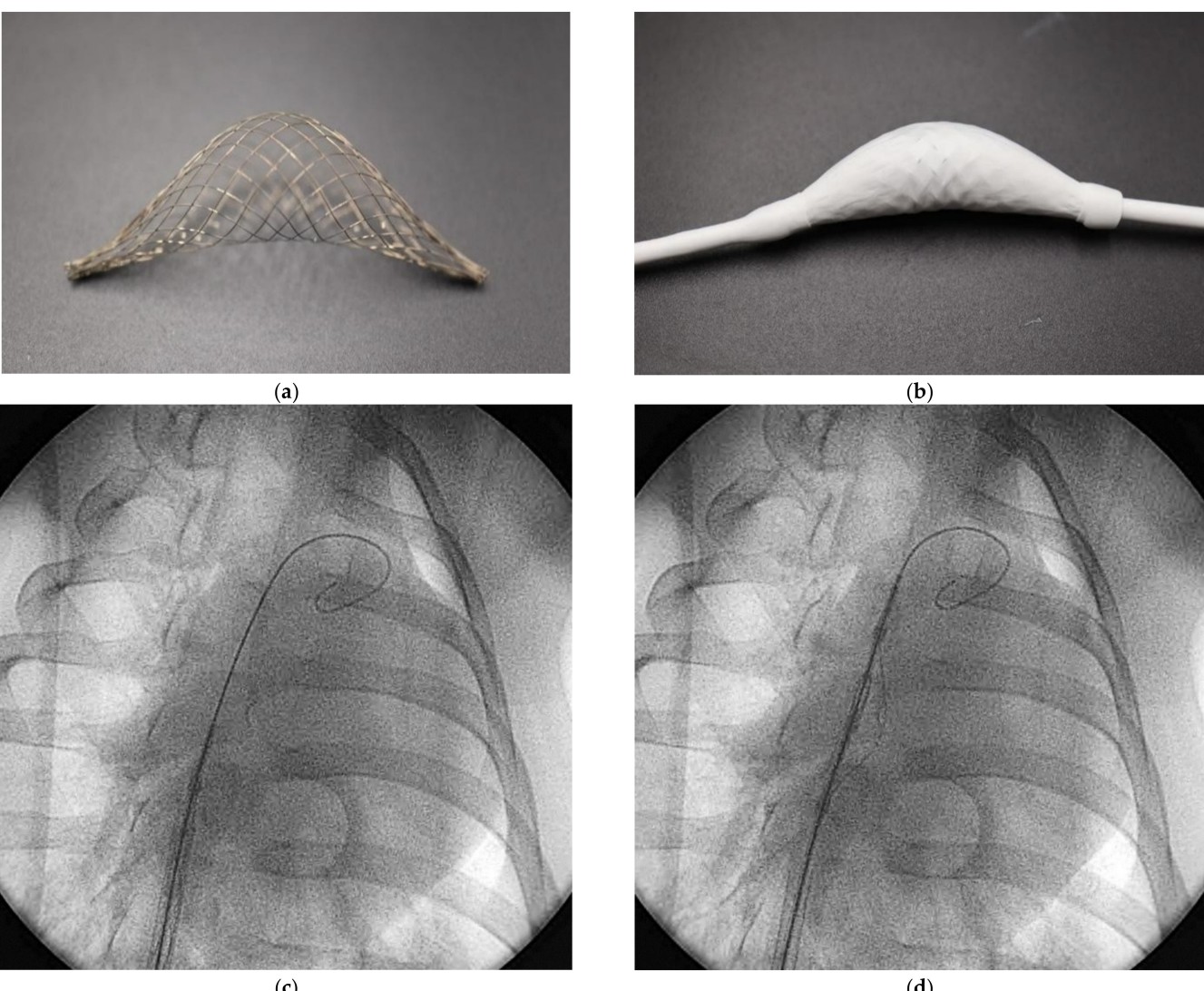

**(a)**    **(b)**

**(c)**    **(d)**

**Figure 1.** Self-expendable mesh spacer. The length of the spacer is 45–50 mm, and the diameter is 12.4 ± 1.5 mm. For its expansion, the inner mesh was pre-shaped (**a**) and then ePTFE-coated (**b**) in advance. The spacer is delivered with the introducer sheath loaded and deployed without additional fluid filler injection. (**c**,**d**).

### 2.1.2. ePTFE Coatings

For biocompatibility (including blood and tissue compatibility) of these spacers, we coated its external surface with ePTFE, widely used as a biocompatible material in artificial blood vessels and valve sectors. Two types of ePTFE (Aeos^TM, Zeus Inc. Orangeburg, SC, USA) were used: a low durometer (Tube type ID 0.128-inch, Wall 0.025-inch, Density $0.35 \pm 0.15$ g/cc) and a high durometer (Ribbon type Sintered Thickness 0.004-inch). They were tested with one or two layers of coating.

### 2.2. In Vivo Test with Swine

This study involved 24 healthy Yorkshire farm swine. The animals were placed under general anesthesia with an anesthesia ventilator (Multiplus, EVD Type, Royal Medical, Pyeongtaek, South Korea). A trained veterinarian carried out anesthesia, paranesthesia, continuous monitoring during the test, and euthanasia procedures. The anesthetized animals were placed in the supine position and their legs were tied to the table using a fabric band to maintain the correct position.

For the procedure of spacer implantation, two experienced interventional cardiologists performed all procedures. A 19Fr sheath (Oscor Inc., Palm Harbor, FL, USA was introduced via the femoral vein and 0.035-inch wire (spring wire, St. Jude medical, St Paul, MN, USA) passed the right atrium (RA) and right ventricle (RV) to the pulmonary artery (PA). The device was collapsed and loaded into the delivery catheter. The catheter loaded with the spacer was delivered through the PA to the RA and the spacer was deployed by retracting the delivery catheter (Figure 1c,d) Our team conducted a preclinical phase to confirm the biocompatibility of the spacer; an experimental environment was established based on the beating hearts of 24 anesthetized swine under identical conditions as the human procedure. The result was confirmed after 6–8 weeks of survival.

### 2.3. Gross and Pathological Evaluation

We technically verified that this spacer device was properly implanted at the target position by X-ray fluoroscopy(Integris H5000F, Philips Medical Systems, Amsterdam, Nederland) and echocardiography(Vivid Q, GE healthcare system, Chicago, Illinois, USA) (Figure 2). After 6–8 weeks, we dissected the heart for inspection to determine any potential side effects and biological interactions between these spacers and surrounding cells and tissues because these spacers were in contact with blood. We also observed whether the shape of the device was well maintained without fillers.

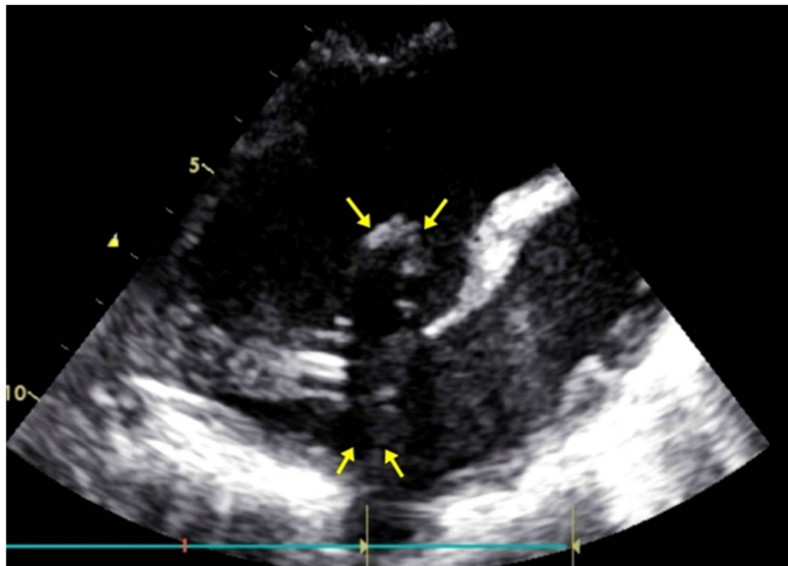

**Figure 2.** Representative case of echocardiography. Echocardiography was used to verify the position of the spacer device.

The devices and tissues were sent to a lab for tissue preparation (Genoss Co., Ltd. Suwon, South Korea). An independent and experienced pathologist evaluated the biocompatibility of spacer and ePTFE coatings. In particular, the composition of endothelial cell formation on the surface of the spacer, cell damage or inflammatory reactions, foreign body reactions, and the components filling inside of spacer were analyzed.

## 3. Results

A total of 24 types of spacer devices were produced, except for two cases suspected of procedural infection. The results were analyzed in the remaining 22 cases.

### 3.1. Self-Expandable Spacer and Procedural Success

A total of 22 types of spacer devices were produced. The ePTFE coating was made in one or two layers without the distinction of its durometer. In all cases, it was well loaded within the delivery sheath and successfully expanded within the heart.

In this swine experiment, all devices can be easily implanted and have shown a good function in self-expansion (Figure 1c,d) within the beating heart. The acute survival rate in the first and mid-term (weeks 6–8) was 100%, and there were no other side effects related to the device and procedure.

### 3.2. Gross Evaluation

As shown in Figure 3a, we confirmed that a smooth cell layer was formed over the ePTFE coating and was well enclosed throughout in the tissue contact area where the edge of the spacer was gently adhered to the surrounding tissue due to natural healing. Thrombosis and damage to surrounding tissues were also not found. In one case, we found that the device located in the pulmonary artery moved by the heartbeat and eventually contact the tricuspid valve leaflet. At this time, the anterior and septal leaflet, which has the most contact with the device, was slightly thicker than the other. In particular, we confirmed that a seroma-like material was formed inside the self-expandable nitinol mesh (Figure 3b).

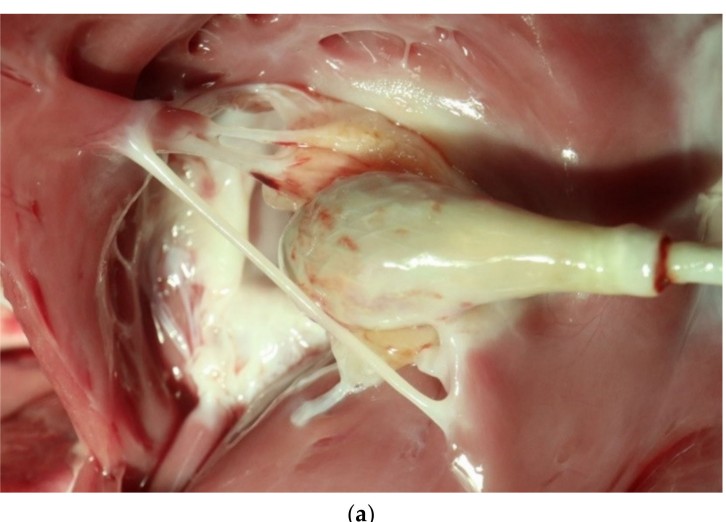
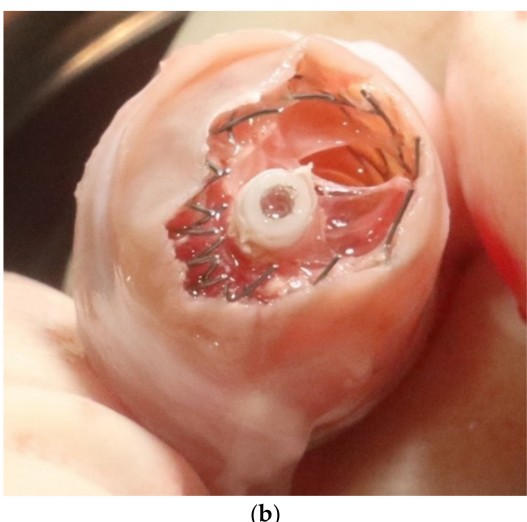

(**a**)                                                (**b**)

**Figure 3.** Gross estimation. (**a**) A smooth cell layer was formed over the ePTFE coating. The anterior leaflet, which has the most contact with the device, was slightly thicker than the other. (**b**) A seroma-like material was formed inside the self-expandable nitinol mesh.

### 3.3. Pathology

We confirmed that neointima covered overall outside the spacer ePTFE coating (Figure 4). We also knew natural healing in which minor inflammation was completely covered with endothelial cells. Some cells can be seen penetrating the ePTFE, but the endothelial cells are well covered on above it, forming a protect film. Although some cells

may have penetrated ePTFE, there was no damage or other abnormality. The spacer is partially opened without being completely sealed, neointima covers not only the exterior but also the inside of the ePTFE coating (Figure 5a). There was no difference depending on the device's ePTFE durometer.

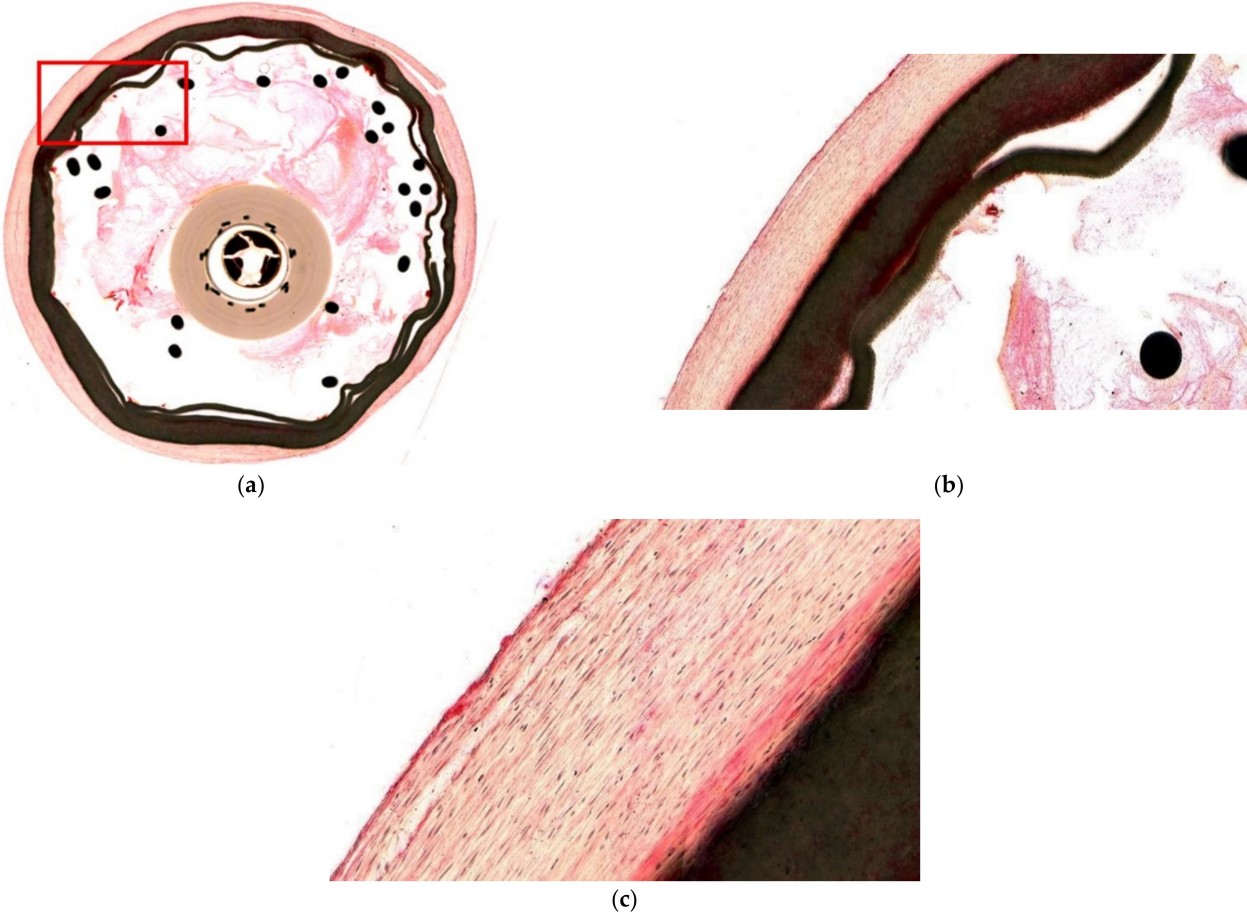

**Figure 4.** Pathology result of closed type spacer. (**a**) Outside the spacer ePTFE coating, epithelialization harbors coverage discontinuously. The space between the outer and inner membrane is filled with fibrin. (**b**) Some of the cells (RBCs) can be seen to be permeated into the ePTFE, but the endothelial cells are well covered above it, forming a kind of protective film. (**c**) The neointima covers the outer membrane without significant foreign body reaction or inflammation.

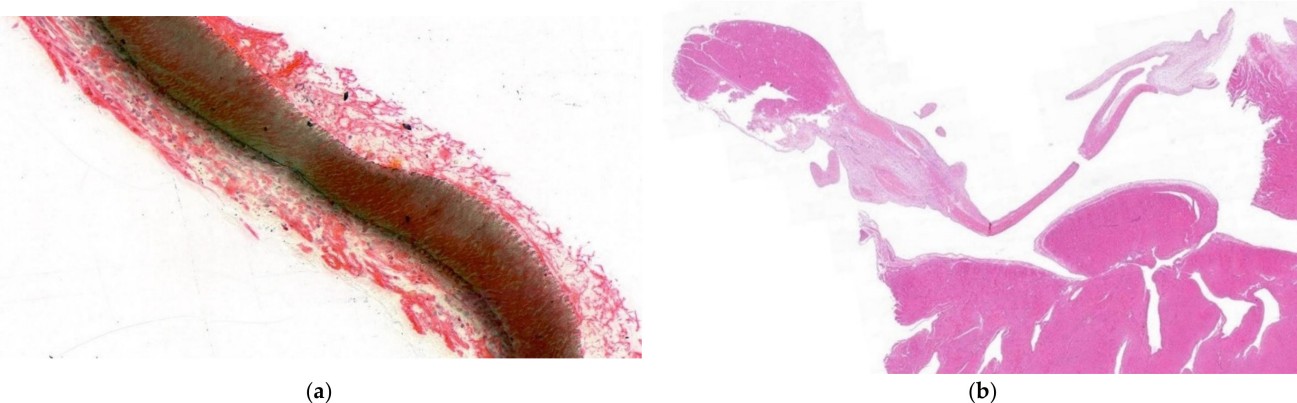

**Figure 5.** Pathology result of the opened type spacer. (**a**) The spacer is not completely sealed and partially opened; the resulting in neointima covers outside and inside the ePTFE coating. (**b**) The leaflet generally shows no symptoms of damage. There is a slight thickening in the edge of the anterior leaflet, which is expected to be a device contact—no noticeable foreign body reaction.

When checking valve tissue that encountered the spacer, there are no indications of damage to the leaflet. The edge of the anterior and septal leaflet is slightly thicker than expected due to the device's contact with the leaflet, which caused mechanical stress (Figure 5b). However, it did not show indications of damage and valve leaflet dysfunction.

## 4. Discussion

This study proposes an ePTFE coated spacer device (made of nitinol self-expandable mesh) as a novel alternative in existing catheter-based cardiac interventional treatments.

The device provides easy delivery with the self-expandable function of the 3D nitinol mesh without any fillers and few possibilities of leakage. We could realize the device as a form of complete implantation in the human body with secured stability as there is no foreign body reaction.

At week 8 of animal testing, we found excellent biocompatibility and anti-thrombogenicity of ePTFE, where the outer surface of ePTFE is completely covered with endothelial cells, and the interior is also filled with fibrin rather than blood clots.

Extensive efforts have been made to develop catheter-based treatments for mitral regurgitation or tricuspid regurgitation. Edward Forma is a spacer device used to treat tricuspid regurgitation. In Forma, a relatively large spacer (up to 20 mm diameter of the long cylindrical body) for effective treatment must be small enough to pass through an introducer sheath (usually less than 10 mm) for transcatheter vascular access. Therefore, the spacer design should allow for inflation or deflation as needed, and the balloon design satisfies this condition. Therefore, most spacer devices have been of the balloon type. However, the balloon design of the spacer also has other challenges, such as spacer filler that is normally designed to be injected outside of the body. The filler should not leak out into the body during long implantable periods. The other is that the device design should have connections with an injection port normally embedded on the subcutaneous tissue of the axilla abdomen. These limitations may include additional safety issues as well as efficacy ones.

Our study showed that the self-expanding spacer with a biocompatible tissue membrane could use blood as a filler material without any concern of leakage. The self-expanding mesh structure made of nitinol allows the collapsible shape to be passed through vascular introducer while expanding in place by the power of self-expanding mesh with the recipient's blood filled in. The body fluid filled in the mesh later becomes an organized material. The ePTFE outer membrane was all covered by healthy nonendothelial tissues. These results show the potential to overcome the limitations of current spacer devices because either 'liquid filler' or 'long connection tail for filler injecting port' may no longer be needed with our new design. We believe that our study result can be applied to a various type of spacer for further development of percutaneous heart valve devices.

## 5. Conclusions

We developed and proposed an ePTFE coated spacer made of nitinol self-expandable mesh that does not require separate fillers for blocking spaces. We showed the biocompatibility of an ePTFE coating on a novel spacer device, through gross inspection and independent pathological analysis by the third party, which circumvents the limitations of the existing filling-type spacer devices. Therefore, this novel nitinol-based self-expanding spacer is expected to be a breakthrough for percutaneous treatment, such as various valve-related diseases and alternatives to the existing filler-type spacer.

**Author Contributions:** Conceptualization, J.-H.K. and M.-K.C.; pathology, D.-H.S.; device management and animal experiments, H.-J.P., writing—original draft preparation and visualization, S.-J.J.; writing—review and editing, M.-K.C.; project administration, J.-H.K. All authors have read and agreed to the published version of the manuscript.

**Funding:** This work was supported by a 2-year research grant from Pusan National University.

**Institutional Review Board Statement:** All animals were handled following the NIH guidelines and Animal Care and Use Committee policies of the Pusan National Yangsan University Hospital (PNUYH, Korea). All animals received humane care. All experimental protocols and studies were approved by the Institutional Review Board (IRB) of the PNUYH [IRB No.:2021-021-A2C0(1)].

**Informed Consent Statement:** Not applicable.

**Data Availability Statement:** Data is contained within the article.

**Conflicts of Interest:** The corresponding author (June-Hong Kim) has intellectual property of the spacer device, stock of Tau-PNU MEDICAL Co., Ltd. and is currently working as clinical director of Tau-PNU MEDICAL Co., Ltd. Su-Jin Jung and Hyeon-Jun Park are currently working in R&D department of Tau-PNU MEDICAL Co., Ltd. The other authors declare no conflict of interest.

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
