# Peer review of "A Percutaneous Catheter Solution as a Spacer for Regurgitant Heart Valve Disease"

_coatings, doi:10.3390/coatings11080926_

Round 1
Reviewer 1 Report
The article under review is describing a novel spacer device design for regurgitant valvulopathies that does not require separate fillers to function. While the new design is interesting, the following aspects of the manuscript need to be addressed:
- In the introduction, the authors write: “To overcome these limitations, many studies have been conducted.” Please provide references for this statement.
- Edits that improve the grammar of the article would be highly beneficial.
- The authors write that animal testing was performed for 8 weeks. Have any attempts been made to quantify the durability of the spacer device after longer periods of time? How about other complications, such as the risk of infection?
- In the methods section, the authors write, “A trained veterinarian carried out all procedures.” However, a sufficient description is not given in regards to what the procedures were.
- Clarify in the manuscript what the economic benefit would be from using the new spacer compared with already available solutions.
Reviewer 2 Report
This study aimed to evaluate the biocompatibility, during 6-8 weeks, of a ePTFE (expanded-Poly tetra-fluoroethylene) device inserted into the free space of the swine heart.
Introduction can be completed by presenting more information about spacer devices.
The authors say that “To overcome these limitations, many studies have been conducted” but they have only 3 references in the whole manuscript. Comparison with results with other studies should be done in the Discussion section.
The authors mentioned that “We technically have verified that this spacer device was properly implanted at the target position by echocardiography”, so results of the echocardiographic evaluation should be also presented.
The material and methods section can be improved by offering more information about the self-expandable mesh implantation procedure.
Round 2
Reviewer 2 Report
The authors did some suggested improvements.
I can understand that further research is necessary for a better comparison, but in the first version of the manuscript, the authors mentioned "To overcome these limitations, many studies have been conducted." and they chose to delete this part instead of providing the required references.
To improve the manuscript the authors should provide more references.
Round 3
Reviewer 2 Report
The authors did the suggested improvements.